# Combined Transcriptomics and Metabolomics Uncover the Potential Mechanism of Plant Growth-Promoting Rhizobacteria on the Regrowth of *Leymus chinensis* After Mowing

**DOI:** 10.3390/ijms26020565

**Published:** 2025-01-10

**Authors:** Ting Yuan, Weibo Ren, Jiatao Zhang, Mohsin Mahmood, Ellen L. Fry, Ru Meng

**Affiliations:** 1Inner Mongolia Key Laboratory of Grassland Ecology and the Candidate State Key Laboratory of Ministry of Science and Technology, Inner Mongolia University, Hohhot 010010, China; yuanting9541@163.com (T.Y.); jtzhang0209@163.com (J.Z.); drmohsin@imu.edu.cn (M.M.);; 2Department of Biology, Edge Hill University, Lancashire L39 4QP, UK

**Keywords:** *Leymus chinensis*, mowing, multi-omics, PGPR, regrowth

## Abstract

Mowing significantly influences nutrient cycling and stimulates metabolic adjustments in plants to promote regrowth. Plant growth-promoting rhizobacteria (PGPR) are crucial for enhancing plant growth, nutrient absorption, and stress resilience; however, whether inoculation with PGPR after mowing can enhance plant regrowth capacity further, as well as its specific regulatory mechanisms, remains unexplored. In this study, PGPR *Pantoea eucalyptus* (B13) was inoculated into mowed *Leymus chinensis* to evaluate its effects on phenotypic traits, root nutrient contents, and hormone levels during the regrowth process and to further explore its role in the regrowth of *L. chinensis* after mowing. The results showed that after mowing, root nutrient and sugar contents decreased significantly, while the signal pathways related to stress hormones were activated. This indicates that after mowing, root resources tend to sacrifice a part of growth and prioritize defense. After mowing, B13 inoculation regulated the plant’s internal hormone balance by reducing the levels and signal of JA, SA, and ABA and upregulated the signal transduction of growth hormones in the root, thus optimizing growth and defense in a mowing environment. Transcriptomic and metabolomic analyses indicated that B13 promoted nutrient uptake and transport in *L. chinensis root*, maintained hormone homeostasis, enhanced metabolic pathways related to carbohydrates, energy, and amino acid metabolism to cope with mowing stress, and promoted root growth and regeneration of shoot. This study reveals the regenerative strategy regulated by B13 in perennial forage grasses, helping optimize resource utilization, increase yield, and enhance grassland stability and resilience.

## 1. Introduction

Plant–microbiome interactions play a key role in many aspects of plant growth promotion, nutrient uptake, and stress tolerance [1]. Certain beneficial bacteria, known as plant growth-promoting rhizobacteria (PGPR), have been identified for their roles in supporting plant growth and stress resilience [2]. These PGPR mobilize nutrients not readily available to plants, such as nitrogen, inorganic phosphate, and iron, by increasing nutrient uptake; promoting nitrogen fixation, solubilization, and mineralization; and improving soil properties to enhance plant tolerance and confer fitness advantages to plant hosts [3]. Recently, most PGPR strains, including Bacillus, Rhizobium, Azospirillum, and Pseudomonas, have been commercialized and widely applied in agriculture to enhance plant growth, mitigate stress responses, and foster sustainable, eco-friendly ecosystems [4,5]. Using PGPR as microbial inoculants offers a cost-effective and eco-friendly approach to support sustainable agricultural practices, enhancing plant growth and increasing crop yields [6].

Understanding the mechanisms of ‘bottom-up’ regulation by which PGPR act on the plant functional genes involved in nutrient absorption, transport, growth, development, and stress resistance is essential for advancing agricultural production. For instance, PGPR can systemically influence transcription levels related to various aspects of growth, nutrient uptake, and metabolism, thus enhancing environmental nutrient absorption and facilitating nitrogen and phosphorus recycling within plant tissues [7]. In addition, beneficial microbes modulate the genes associated with carbohydrate metabolism, nutrient and sugar uptake, transport, and secondary metabolite production, ultimately promoting plant growth [8]. Inoculation with PGPR regulates plant growth and enhances resistance in the family Poaceae, such as maize [3], wheat [9], sugarcane, and rice [10]. Although many studies have shown that well-characterized PGPR have been successfully employed in agriculture to directly enhance plant nutrient absorption, growth, yield, crop stress resistance, and modulate phytohormone levels [11,12], few have focused on the effect of PGPR on grassland plant *Leymus chinensis*.

Grassland ecosystems have served as an essential component of terrestrial systems, widely utilized for livestock production, with significant production and ecological importance [13]. As a common grassland utilization measure worldwide, in addition to grazing, mowing is a key driver of nutrient cycling and plant diversity in grassland ecosystems and may affect ecosystem function in grasslands [14,15]. Moderate mowing may increase the compensatory growth of plants, promote plant photosynthetic efficiency and grassland productivity, and be favorable to forage yield and quality [16]. Numerous studies have also shown that short-term mowing can stimulate nutrient cycling in grassland ecosystems, improve plant nutrient absorption, and increase community productivity [17,18]; however, long-term heavy mowing may reduce plant regrowth capacity and soil fertility, resulting in stunted plant growth and reduced aboveground biomass, plant diversity, and productivity [19]. Mowing also weakens the dominance of *L*. *chinensis*, leading to shifts in plant community composition [20].

Mowing impacts the photosynthetic organs of plants, disrupting the original source–sink relationships and the carbon/nitrogen (C/N) balance across different plant organs, which, in turn, affects grassland vegetation [21]. To meet the demand for shoot nutrients (N and P) after mowing, plants absorb nutrients from the soil and circulate them internally by absorbing and redistributing storage organs (such as roots), allowing the nutrients to be re-used for shoot growth [22]. As the primary interface between plants and soil, roots are the main organ of nutrient absorption, transformation, circulation, and transport to shoots [23]. During mowing, plant roots are not directly damaged but are indirectly affected by the soil properties and the physiological changes in shoots caused by mowing [24]. Several reports have described changes in leaf area and dry biomass in response to mowing but did not alter root traits within and between species [25,26]. While numerous studies have examined the effects of mowing on shoot traits and nutrient dynamics in grassland ecosystems [27,28], there has been limited focus on how mowing impacts root nutrient accumulation and growth during plant regrowth.

Carbohydrates are the primary products of photosynthesis that supply essential energy for plant growth. When plants encounter various biotic and abiotic stresses, such as mechanical injury, drought, or herbivory, they adapt to these environmental challenges by catabolizing carbohydrates to release energy, which may lead to changes in their morphological and physiological characteristics [29]. Starch and other polysaccharides are converted into fructose and glucose, providing the necessary energy and carbon skeletons for plant regrowth [30]. Nitrogen metabolism, in turn, supplies the nitrogen framework required for amino acid synthesis and metabolism following mechanical damage [31]. Additionally, the regrowth of forage plants after mowing involves repairing photosynthetic organs, primarily through leaf growth and expansion, a process regulated by sugar and hormonal signals. In response to environmental stresses, plants reallocate resources between growth and defense. Stress hormones such as abscisic acid (ABA), jasmonic acid (JA), and salicylic acid (SA) play significant roles in regrowth after injury by regulating the division rate of organizer cells, thus contributing to plant recovery and defense mechanisms [31,32].

The ability of grassland plants to regrow after grazing or mowing is essential. *L. chinensis*, a key forage grass and dominant perennial species in the Inner Mongolian grasslands of eastern Eurasia, exhibits strong adaptability to abiotic and biotic stresses [33,34]. The regrowth of this dominant species is crucial for supporting grassland productivity, maintaining community stability, and ensuring grassland sustainability. While studies on grassland plant regrowth have primarily focused on biomass, regrowth rate, aboveground vegetation, and soil physicochemical properties, the impact of combined PGPR inoculation and mowing on *L. chinensis* regrowth remains unexplored.

*L. chinensis* was the research focus and studied using combined transcriptomics and metabolomes. The selected PGPR strain *P. eucalyptus* B13 was used to inoculate *L. chinensis* based on the principal growth-promoting properties (phosphate solubilization, nitrogenase activity, and IAA production) (unpublished data). This research aimed to explore how PGPR *P. eucalyptus* B13 inoculation regulates changes in root physiology, genes, and metabolites and further promotes the regrowth of *L. chinensis* shoots after mowing.

## 2. Results

### 2.1. Isolation and Identification of B13

Strain B13 was isolated from *L. chinensis* rhizosphere soils in degraded grassland due to long-term overgrazing of the Inner Mongolia Grassland Ecosystem Research Station (43°38′30″ N, 116°42′20″ E) in northern China. The sequencing results of B13 were analyzed using BLAST from NCBI, and a phylogenetic tree was constructed using the 16S rRNA gene sequences with MEGA 11, which showed that B13 has the closest genetic relationship with *P. eucalypti* (Figure 1). The results of the growth-promoting characteristic analyses showed that the indole-3-acetic acid (IAA) production, dissolved inorganic phosphorus, and nitrogenase activity of isolated *P. eucalypti* B13 were 29.41 μg/mL, 107.38 μg/mL, and 186.38 IU/L, respectively (Appendix A), indicating high IAA production, inorganic phosphate-releasing capacity, and N-fixation capacity.

### 2.2. Effects of B13 Inoculation on Growth of L. chinensis

Under mowing, after 7 days of B13 inoculation, the height of *L. chinensis* was significantly higher than the control (Figure 2A). At 14 days after B13 inoculation, the height was significantly higher than the control, and the height of inoculation after mowing was significantly higher than the non-mowing group. Moreover, this trend persisted over time. After 21 days, mowing before inoculation and inoculation after mowing increased the regrowth height of *L. chinensis* significantly compared with the control (Figure 2B), and the regrowth height of B13 inoculation after mowing was the most significant. After 35 days, mowing increased the plant leaf width, height, and shoot fresh weight of *L. chinensis* significantly but had no significant impact on stem diameter and root fresh weight (Figure 2). After B13 inoculation, the plant height, leaf width, stem width, shoot fresh weight, and root fresh weight were increased significantly by 14.07%, 21.52%, 23%, 41.76%, and 30.41%, respectively. In addition, B13 inoculation after mowing had a more significant promoting effect on the above phenotypic traits, which increased by 27.60%, 37.53%, 21.82%, 95.22%, and 49.90%, respectively (Figure 2).

### 2.3. Effects of B13 Inoculation on the Accumulation of Nutrients and Sugars by L. chinensis Root

The accumulation of total nitrogen (TN), total carbon (TC), total potassium (TK), total phosphorus (TP), soluble sugar, glucose, fructose, and sucrose in *L. chinensis* root was significantly reduced after mowing (Figure 3A–D); however, the content of TP, TK, soluble sugar, and glucose was increased significantly by 20.17%, 9.36%, 15.5%, and 39.87%, respectively, after B13 inoculation (Figure 3C,D,F,I). In addition, inoculation with B13 after mowing resulted in a gradual recovery of these nutrient and sugar contents and significantly increased the contents of TN, TP, soluble sugar, glucose, fructose, and sucrose in *L. chinensis* roots by 5.61%, 33.98%, 11.85%, 14.90%, 21.26%, and 15.27%, respectively, compared with the control (Figure 3B,C,F–I).

### 2.4. Effects of B13 Inoculation on the Phytohormones and Antioxidant Enzymes in L. chinensis

The contents of the plant hormones IAA, zeatin (tZ), JA, SA, and ABA increased significantly after mowing, and the contents of IAA and SA increased significantly after B13 inoculation (Figure 4A,E). Inoculation with B13 after mowing resulted in the contents of IAA and tZ increasing significantly, but the contents of gibberellin (GA3), JA, and SA decreased significantly (Figure 4A–E). For the antioxidant enzymes, the peroxidase (POD), polyphenol oxidase (PPO), catalase (CAT), and lipoxygenase (LOX) activities increased significantly after mowing. In contrast, the superoxide dismutase (SOD) and malondialdehyde (MDA) activities decreased significantly after mowing (Figure 4G–L). There was no significant change in these enzymes’ activity after inoculation with B13, while inoculation with B13 after mowing resulted in the activities of SOD and MDA decreasing significantly. These results suggested that the plant antioxidant defense system might be activated after mowing; however, at the same time, B13 inoculation after mowing alleviated the peroxidation damage of roots.

### 2.5. Transcriptomic Analysis

The roots of *L. chinensis* inoculated with strain B13 were used for transcriptomic sequencing. A total of 385,044,932 clean reads was obtained for 12 cDNA libraries, with an average of 78,580,123 reads per sample; the average GC ratio was 54%, Q20 exceeded 97%, and Q30 exceeded 92% (Appendix A), indicating the high quality of these sequencing data. In addition, differentially expressed genes (DEGs) (threshold: Padj < 0.05; |log2FC| > 1) between the non-inoculated controls and inoculated samples found 5304 DEGs were regulated differently, among which 1487 were upregulated and 3817 were downregulated (Figure 5A). After mowing, compared with the non-inoculated controls, B13 inoculation altered the expression of 13,970 DEGs significantly, among which 7020 were upregulated and 6950 were downregulated (Figure 5A). For B13 inoculation after mowing, 37,271 DEGs were significantly regulated, among which 1487 were upregulated, and 23,833 were downregulated (Figure 5A). Between the non-mowed controls and mowed samples, 27,433 DEGs were regulated differently, among which 9658 were upregulated, and 17,775 were downregulated (Figure 5A).

The analysis of the Venn diagrams of the DEGs among the different comparison combinations showed that there were 360 common DEGs that were regulated (Figure 5B). Moreover, a total of 3850 genes were co-expressed in RCK vs. R13 and RCK vs. MR13. The DEGs were clustered into eight significant expression profiles (Figure 5C). Profile 3 included 4058 genes upregulated significantly after mowing or/and inoculation with B13. Under mowing, Profile 5 included 2759 rapidly upregulated genes after B13 inoculation. In Profile 11, 3888 DEGs were upregulated after B13 inoculation but downregulated under mowing and B13 inoculation simultaneously. Nevertheless, most of these genes were downregulated significantly after mowing (Profiles 1 and 2). Among the top 10 most enriched Kyoto Encyclopedia of Genes and Genomes (KEGG) pathways in each comparison group, phenylpropanoid biosynthesis, nitrogen metabolism, and plant hormone signal transduction were the most significantly enriched pathways (Figure 5D–G). These findings suggest that these pathways play crucial roles as key metabolic processes in *L. chinensis* in response to mowing or B13 inoculation.

### 2.6. B13 Inoculation Affected Plant Hormone Signal Transduction Pathway

The genes associated with plant hormone signal transduction were analyzed. The results showed significant alterations in the expression levels of the genes related to IAA, ABA, SA, cytokinin (CTK), ethylene (ETH), GA3, brassinosteroid (BR), and JA (Figure 6). In the IAA pathway, the genes encoding auxin positive regulatory response factor (AUX/IAA) and auxin response factor (ARF) were mostly upregulated in both B13-inoculated and non-inoculated *L. chinensis* after mowing, but to a greater extent in the presence of inoculum. In addition, there were four SAUR genes that were upregulated after inoculation with B13 but downregulated in the other comparison groups. In the CTK pathway, two histidine-phosphate transporter (AHP) genes were upregulated in all the comparison groups, and two A-ARR genes were upregulated after mowing and B13 inoculation under mowing. In the GA pathway, the genes encoding DELLA and TFs were upregulated in both B13-inoculated and non-inoculated *L. chinensis* after mowing. Among the ABA and ETH pathways, the related genes encoding enzymes such as protein phosphatase 2C (PP2C), serine/threonine-protein kinase SRK2 (Snrk2), ethylene receptor (ETR), and ethylene insensitive 2 (EIN2) were mostly upregulated in all the comparison groups, while the genes encoding abscisic acid receptor (PYR/PYL), mitogen-activated protein kinase (MPK6), and ethylene-responsive factors (ERF1/2) were downregulated. In addition, ABA-responsive element binding factor (ABF) genes were upregulated after mowing but downregulated in the presence of inoculum. In the JA pathway, the gene encoding the F-box protein receptor (COI1) was upregulated in all the comparison groups. The gene encoding jasmonic acid-amino synthetase (JAR1) was upregulated after B13 inoculation but downregulated after mowing and with B13 inoculation after mowing. Moreover, three jasmonate ZIM domain-containing protein (JAZ) genes were upregulated under mowing but downregulated under B13 inoculation. In the SA pathway, most genes encoding nonexpresser of pathogenesis-related genes 1 (NPR1) were upregulated after mowing but downregulated in the presence of inoculum. In addition, most TGA transcription factor (TGA) genes were upregulated only after B13 inoculation.

### 2.7. B13 Inoculation Affected Genes Related to Nutrient and Sugar Transport

The expression levels of four nitrate transporter-related genes (NPFs) were largely upregulated by B13 inoculation and inoculation after mowing. In comparison, five NPFs were upregulated under mowing and inoculation with B13 after mowing (Appendix A). Most phosphate transport-related genes were upregulated in all the comparison groups but to a greater extent in the presence of inoculum after mowing (Appendix A). Further, the expression levels of most sugar-related genes were upregulated after B13 inoculation in both the mowing and non-mowing groups (Appendix A).

### 2.8. Characterization of Plant Root Metabolome in Response to B13 Inoculation

There were 124 differentially expressed metabolites (DEMs) (99 up and 4 down), 47 DEMs (47 up and 0 down), 72 DEMs (99 up and 4 down), and 93 DEMs (89 up and 4 down) that were detected in the RCK vs. MCK, RCK vs. R13, MCK vs. MR13, and RCK vs. MR13 comparisons, respectively (VIP > 1.0, *p*-value < 0.05, FC ≥ 1, or FC ≤ −1) (Figure 7A). The analysis of the Venn diagrams of the DEMs among the different comparison groups showed 28 commonly regulated DEMs (Figure 7B). Further, 12 and 14 DEMs were present only after inoculation and mowing compared with the control, respectively. All the DEMs were classified into amino acids (58 compounds), sugars (11 compounds), organic acids (11 compounds), lipids (11 compounds), organoheterocyclic (10 compounds), flavonoids (5 compounds), nucleotides (4 compounds), phenolic acids (3 compounds), vitamins (3 compounds), phytohormones (2 compounds), and others (6 compounds) in eleven categories (Figure 7C). Compared with the other comparison groups, more metabolites were detected in the RCK vs. MRCK and RCK vs. MR13 comparison groups and amino acids were the most abundant metabolites, followed by sugars, organic acids, and lipids. Almost all the DEMs were significantly enriched under mowing, with B13 inoculation, and with inoculation after mowing, among which the number of DEMs produced under mowing was the highest (Figure 7C,D).

### 2.9. Integrated Metabolomic and Transcriptomic Analysis

In order to identify potentially related pathways under B13 inoculation in response to mowing stress, a KEGG enrichment analysis with a *p*-value histogram was performed. The results showed that 10 differentially significant metabolic pathways were annotated in all the comparison groups. In the comparison of RCK vs. R13 and MRCK vs. MR13, both DEMs and DEGs were enriched significantly in the pathways for alanine, aspartate, glutamate, nitrogen, and butanoate metabolism (Appendix A). Based on the results of the transcriptome and metabolome association analysis, the relevant genes and metabolites involved in these metabolism pathways were used to map the metabolic pathways (Figure 8).

In alanine, aspartate, and glutamate metabolism, 8 DAMs and 163 DEGs were enriched significantly across the four comparison groups, most of which are linked to the tricarboxylic acid (TCA) cycle and amino acid metabolism pathway. The majority of the genes related to various enzymes, including argininosuccinate synthase (argG) and glutamine--fructose-6-phosphate transaminase (glmS), were consistently upregulated in RCK vs. R13, RCK vs. MRCK, and MRCK vs. MR13. In addition, most genes encoding omega-amidase (NIT2), glutamate decarboxylase (GAD), and 4-aminobutyrate aminotransferase (ABAT) were consistently upregulated in MRCK vs. MR13, RCK vs. MRCK, and RCK vs. MR13. Particularly, most genes encoding aspartate aminotransferase (GOT1), glutamate synthase (GLT1), RHH-type transcriptional regulator (putA), aspartate carbamoyltransferase (CAD), and succinate-semialdehyde dehydrogenase (SSADH) were upregulated in all the comparison groups, which enhanced biosynthesis for aspartate- and glutamate-derived pathways; however, asparagine synthase (asnB) was downregulated in R13 vs. RCK, MRCK vs. RCK, and MR13 vs. MRCK. Consistently, DAMs, including L-aspartic acid (L-asp), L-asparagine (L-asn), and citric acid, were upregulated significantly after inoculation with B13, mowing, and inoculation after mowing. Increased levels of these DAMs suggest higher nitrogen assimilation and carbon flow through the TCA cycle, which supports energy production and the synthesis of amino acids needed for growth and repair.

In the nitrogen metabolism pathway, inoculation with B13 downregulated the expression levels of nitrate reductase (NR) and nitrite reductase (NiR) in both the mowing and non-mowing groups; however, the majority of genes, including carbonic anhydrase (CA), cyanate lyase (cynS), and glutamine synthetase (glnA), were upregulated in RCK vs. R13, RCK vs. MRCK, and MRCK vs. MR13. In particular, nitrate transport genes (NRT) and glnA were mainly upregulated under mowing or inoculation after mowing. Meanwhile, the metabolomics results show that L-glutamic acid was upregulated under B13 inoculation, and L-glutamine was upregulated in all the comparison groups. The upregulation of L-glutamic acid and L-glutamine demonstrates an enhanced ability to assimilate nitrogen into amino acids, which are crucial for growth and repair.

Among the butanoate metabolism pathway, inoculation with B13 upregulated the most gene expression levels including glyoxylate/succinic semialdehyde reductase (GLYR), hydroxymethylglutaryl-CoA (HMGCS), and synthase and acetyl-CoA C-acetyltransferase (ACAT) in MRCK vs. MR13, RCK vs. MRCK, and MRCK vs. MR13. In contrast, enoyl-CoA hydratase (paaF) was downregulated. Moreover, the metabolite abundances of 4-aminobutyric acid, succinic acid, L(-)-malic acid, and maleic acid were upregulated after mowing, inoculation, and inoculation after mowing. These accumulated metabolites feed into the TCA cycle, ensuring that the plant maintains ATP production and metabolic homeostasis under stressful conditions.

In the starch and sucrose metabolism pathways, most genes, including insoluble isoenzyme (INV), sucrose synthase (SUS), glucose-6-phosphate isomerase (GPI), 1,4-alpha-glucan branching enzyme (GBE1), UTP-glucose-1-phosphate uridylyltransferase (UGP2), phosphoglucomutase (pgm), maltase-glucoamylase (MGAM), glucose-1-phosphate adenylyltransferase (glgC), and alpha-amylase (AMY), were upregulated in RCK vs. R13, RCK vs. MRCK, and MRCK vs. MR13. In contrast, the expression levels of the majority of the genes in RCK vs. MR13 were higher than in the other two groups. Especially, most genes encoding glucanendo-1,3- beta-D-glucosidase (EGLC), glycogen phosphorylase (PYG), isoamylase (ISA), hexokinases (HK), and glycogen synthase (GYS) were upregulated in all the comparison groups. The mobilization of starch reserves and sugar metabolism ensures sustained energy supply during stress recovery, promoting resilience and regrowth after mowing.

To further explore the relationship between DAMs and DEGs in *L. chinensis* with B13 inoculation after mowing, a co-expression network analysis of the metabolome and transcriptome was conducted. From the Venn diagram of the transcriptome and metabolome data, 360 DEGs and 28 DEM co-enriched by different treatments were selected, and a Pearson correlation coefficient > 0.95 and *p*-value < 0.05 were set as the threshold values for the co-expression network analysis. The numerous DEMs strongly associated with these DEGs were Ala-Gly, D-glutamine, and L-lysine, followed by glutamine and lysine (Appendix A). Further, the co-expression networks of DEGs and DEMs in key metabolic pathways identified by transcriptome and metabolome association were also analyzed. These results suggested that seven DEMs were highly positively linked with DEGs, especially maleic acid and L-asn (Appendix A), similar to the above results. The accumulation of related metabolites may, therefore, either directly or indirectly control changes in these genes.

### 2.10. qRT-PCR Analysis

To verify the accuracy of the transcriptome sequencing data, nine DEGs related to nitrate transport; phosphate transport; sugar transport; starch and sucrose metabolism; alanine, aspartate, and glutamate metabolism; and plant hormone signal transduction were analyzed using qRT-PCR for expression validation. The results indicated that the expression trends of the seven DEGs across the four treatments generally matched those observed in the transcriptome sequencing data (Appendix A), confirming that these transcriptome data are accurate and reliable.

## 3. Discussion

### 3.1. B13 Inoculation Promotes Regrowth of L. chinensis After Mowing by Enhancing the Absorption and Transport of Nutrients

The reallocation of nutrients from roots to shoots is an important nutrient-use strategy for plant regrowth in grasslands [35]. Particularly in grasslands, plants adapt to herbivory or mowing through shoot regrowth, in which nutrient absorption and reallocation processes play a key role [36]. In the process of plant regrowth, 48–97% and 58–79% of the nitrogen and phosphorus were derived from root-stored nutrients, indicating that the root redistribution of nitrogen and phosphorus was an important nutrient acquisition process to maintain plant regrowth [37]. This study showed that PGPR B13 inoculation significantly increased plant height, stem width, leaf width, shoot fresh weight, and root fresh weight in both mowed and non-mowed *L. chinensis* (Figure 2), consistent with the findings from previous research [38]. PGPR enhance plant growth through both direct and indirect pathways, including phosphorus solubilization, N fixation, siderophore production, volatile compound release, and the induction of systemic resistance. Additionally, PGPR can regulate the expression of plant growth and development-related genes by producing hormones and mobilizing nutrients for easy uptake by plants and increasing plant nutrient use efficiency [39]. Notably, the accumulation of TC, TN, TP, TK, soluble sugar, glucose, fructose, and sucrose was significantly reduced after mowing. At the same time, the contents were significantly increased except for TK in the B13 inoculation with mowing treatment (Figure 3). In addition, nitrate transporter genes, phosphate transport genes, and sugar transporter genes were strongly upregulated after B13 inoculation both in mowing and non-mowing and inoculation under mowing conditions, which upregulated the expression of these transport genes further (Appendix A). The highest degree of upregulation was observed with inoculation following mowing, suggesting that these genes may respond to systemic signals circulating between the roots and shoots of inoculated plants, thus playing a critical role in promoting plant growth. Numerous studies have shown that PGPR inoculation significantly influences plant growth, nutrient uptake, translocation, and metabolite regulation [12,40]. These findings show that B13 inoculation can notably enhance growth and nutrient acquisition in host plants, particularly in mowed *L. chinensis*. This aligns with previous research indicating that applying PGPR boosts nitrogen uptake, transport, and metabolism in roots, with significant upregulation of the NRT gene family [41,42]. This indicates that plant roots mobilize nutrients for shoot regrowth under mowing, and B13 inoculation promotes the uptake and transport of root nutrients under mowing to promote root growth and shoot regrowth. The results demonstrate that B13 application improved nutrient uptake by roots, modified nutrient acquisition and metabolism pathways, and provided adequate nutrient support for shoot regeneration [43].

### 3.2. B13 Inoculation Responds to Mowing L. chinensis by Regulating Plant Hormone Signaling Pathways

Plant hormones and signaling transduction pathways regulate plant growth in response to abiotic stress and microbial symbiosis [44]. Mowing can trigger changes in plant hormone signaling pathways, causing plants to prioritize defense against stress and regulate resource allocation [45]. The addition of PGPR can further optimize gene expression, enhance growth, improve resistance, and promote resource absorption. In *L. chinensis*, regrowth after mowing includes both leaf repair and active shoot growth, with minimal impacts on root growth. During this period, the levels of IAA and CTKs in roots increase (Figure 3A,B), likely benefiting leaf regrowth as these hormones are transported to the shoots to aid leaf repair, consistent with previous findings [45]. It has been reported that IAA and CTKs can promote the regrowth of winter wheat and barley after mowing [46]. After mowing, plants redistribute auxin to the roots, and auxin signals promote root growth and shoot recovery by promoting the elongation and division of root cells [47]. CTK mainly accumulates in the root tip to promote cell division and differentiation and support rapid root growth, providing solid root support for shoot regeneration [46]. Further, the contents of IAA and CTKs were further increased by B13 inoculation after mowing (Figure 4A,B). With regard to auxin signaling, under mowing, the genes encoding AUX/IAA and ARF were upregulated in both the B13-inoculated and non-inoculated treatments, but to a greater extent in the presence of inoculum (Figure 6). In addition, the gene expression level of GH3 was downregulated after mowing, but upregulated in the presence of inoculum. The activation of these genes increased the auxin response pathway significantly, causing rapid root cell division and elongation. The results indicate that B13 regulated the root response to mowing by reprogramming auxin metabolism and activating auxin signaling [48], potentially through bacterial signaling molecules that boost auxin transport and accumulation in roots.

In the CTK pathway, A-ARR and B-ARR are key receptors that transmit signals by modifying the phosphorylation state [49]. Under mowing, the results confirmed that the gene expression levels of cytokinin response 1 (CRE1) and histidine-phosphate transporter (AHP) were upregulated, and the negative regulator A-ARR was downregulated (Figure 6); however, in the presence of inoculum after mowing, the expression of the ARR genes was reduced further, and the expression of CRE1 was increased further, consistent with the increase in tZ content (Figure 4B). This indicates that inoculation with B13 significantly increased CTK signaling in *L. chinensis* roots after mowing, thus promoting root growth and shoot regrowth. These results suggest that after mowing, auxin and CTK promote shoot recovery through redistribution and synergistic action [50]. With PGPR inoculation, the signaling of these hormones is enhanced further, particularly with regard to the accumulation of auxin and CTK, significantly boosting root growth and regeneration efficiency, and thus further supporting shoot recovery after mowing [46].

Plants adapt to environmental challenges by reallocating resources between growth and defense mechanisms. The response pathways of stress-inducing hormones JA, SA, ABA, and ETH interact with growth signaling pathways, ultimately influencing plant growth [51]. Mowing can activate the expression of defense-related genes in roots but may result in some resources being sacrificed, resulting in root growth inhibition. Mowing enhanced the concentrations of JA, SA, and ABA, but these concentrations decreased in the presence of inoculum (Figure 4D–F), suggesting a shift in the growth and defense strategy of *L. chinensis* in the presence of inoculum after mowing. This is also illustrated by the activities of antioxidant enzymes (Figure 4D–I). After mowing, the expression levels of the genes encoding JAZ and MYC2 were upregulated but became downregulated in the presence of inoculum (Figure 6). JA-Ile is a crucial compound in the JA signal transduction pathway; however, the downregulation of JAR1 gene expression after inoculation under mowing conditions suggests an inhibition of the reversible conversion between JA-Ile and JA [52]. Consequently, the downregulation of MYC2 and most JAZ genes may reduce the capacity for protein interactions that regulate JA responses [53,54]. These results indicate that B13 inoculation optimizes whole plant adaptation to mowing stress.

It is well established that ABA and ETH play key roles in enhancing abiotic stress tolerance in *L. chinensis* [55]. Under mowing, the expression levels of most genes related to PP2C, SnRK2, and ABF were upregulated, indicating the activation of the ABA signaling pathway (Figure 6). This aligns with previous studies suggesting that the upregulation of PYL may lead to the increased expression of PP2C, thus activating SnRK2 and ABF to initiate defense mechanisms further [56]. Additionally, B13 inoculation downregulated the gene expression levels of the PYR/PYL family genes, thus reducing the capacity to interact with ABA. Further, the expression of ABA-induced positive regulator ABF was also inhibited, indicating that the ABA signaling pathway may be inhibited. In the SA signaling pathway, most genes encoding NPR1 were strongly upregulated under mowing, and inhibited significantly in the presence of inoculum. Notably, the differential regulation of these related transcripts suggests that ABA and SA signaling may be involved in the establishment of bacteria after inoculation in mowed *L. chinensis* [57,58].

The results confirmed that in regulating gene expression after mowing, the PGPR B13 helps plants strike a balance between the defense signaling (JA, SA, and ABA) pathway and growth signaling (auxin and CTK) pathway; enhances root absorption, structure, and resistance; and significantly promotes plant regeneration and overall recovery in mowed *L. chinensis*. This may provide the basis for phytohormone signals to be involved in the regrowth of *L. chinensis* in more complex ways, such as crosstalk between multiple hormones and growth and defense tradeoffs [57,59]. In short, the results indicated that B13 participates in the pathway of plant signal transduction and may play a key role in mowing stress adaptation and plant regrowth. These hormone interactions regulate the growth–defense tradeoff and further affect the regrowth of *L. chinensis* inoculated with B13 after mowing.

### 3.3. B13 Inoculation Promotes Root Growth and Regrowth of L. chinensis After Mowing Through Amino Acid Metabolism

Amino acid metabolism pathways such as alanine, aspartate, and glutamate are essential for energy supply, nitrogen metabolism, and stress responses in plants, which were significantly regulated after mowing or inoculation (Figure 8A). L-asp represents a critical metabolite hub linked to a variety of metabolic pathways and plays a key role in plant nutrient absorption, energy supply, growth and development, and stress response. L-asp was highly accumulated owing to inoculation or mowing stress, aligning with previous findings [60]. Consistently, the abundance of L-asn increased in all the comparisons, especially under B13 inoculation after mowing. AsnB, encoding an asparagine synthetase, catalyzes the L-asp conversion to asparagine based on aspartic acid [60]. The results indicated that the asnB genes were downregulated significantly after inoculation with B13 under both the mowing and non-mowing treatments; therefore, the degradation of L-asp was inhibited, suggesting a significant accumulation of L-asp at the transcriptional level after inoculation with B13 under both the mowing and non-mowing conditions [61].

γ-Aminobutyric acid (GABA), a nonproteinogenic amino acid, accumulates widely in many plant organs and tissues, serving as both a metabolite and signaling molecule involved in plant growth, development, and responses to various environmental stresses. Glu, as a nitrogen donor for amino acid synthesis, can be converted to GABA by GAD and glutamate dehydrogenase (GDH), respectively. This process plays a crucial role in balancing plant C/N metabolism and regulating plant growth, development, and defense responses [62]. GABA serves as both a metabolite and signaling molecule involved in plant growth and responses to various environmental stresses [63]. In this study, the expression levels of most genes coding for GAD (a main enzyme in GABA biosynthesis) were upregulated, and GABA accumulation was increased after mowing, which may help plants cope with oxidative stress caused by mowing. Furthermore, the expression levels of most genes involved in GABA synthesis, as well as the accumulation of related metabolites, were higher after simultaneous B13 inoculation following mowing compared with mowing alone, aligning with the accumulation of glutamate and the expression levels of GAD in response to mowing stress, contributing to the regulation of plant growth [64]. Simultaneously, previous studies have also confirmed that the rapid accumulation of GABA in the rhizosphere of tomatoes inoculated with PGPB in response to abiotic stress further regulates plant growth and development [60,65,66]. Additionally, GABA alleviates salt damage and enhances growth in tomato and rice plants by modulating amino acid synthesis, the TCA cycle, and altering the expression of the GAD gene [67,68]. PGPR can help plants recover faster by coordinating the metabolic cycle between alanine, aspartate, and glutamate, optimizing the nitrogen metabolism and carbon metabolism of plants after mowing. By regulating these genes, PGPR promotes the overall growth and stress adaptation of plants after mowing.

### 3.4. B13 Inoculation Altered Carbohydrate Metabolism and Energy Metabolism of L. chinensis Root After Mowing

As the main product of photosynthesis, carbohydrate metabolism provides essential energy and carbon skeletons for plant growth and stress resistance. During the recovery period after mowing, plants often need rapid carbohydrate synthesis to promote regeneration, and PGPR can help plants recover by promoting photosynthesis and carbohydrate metabolism, enhancing sugar synthesis, and speeding the recovery process [69]. In *L. chinensis* roots, starch and sucrose metabolism are the principal carbohydrate metabolic pathways, occurring via two distinct routes [70]. Starch aids plants in withstanding external stresses, with stored starch offering resources for recovery following hydration. During starch synthesis, fructose is first converted to fructose-6-phosphate (Fruc-6P) by the activities of HK, then further transformed to glucose-6-phosphate (Gluc-6P) and glucose-1-phosphate (Gluc-1P) through the actions of GPI and pgm. Finally, Gluc-1P is further converted to starch by the activities of glgC, glgA, and GBE1 [71]. In the present study, most of the genes encoding HK, GPI, pgm, glgA, and GBE1 were upregulated significantly in RCK vs. R13, RCK vs. MRCK, and RCK vs. MR13 (Figure 8D). Additionally, the gene expression levels in RCK vs. MR13 were higher than in the other two groups, suggesting that strain B13 enhances ADP–glucose conversion to produce two types of starch through glgA regulation after mowing, thus accelerating starch accumulation in *L. chinensis* roots and promoting root growth and development [72].

Sucrose is the main form of photosynthate transport in plants. In the process of sucrose synthesis and decomposition, triose phosphate is transformed into sucrose by SPS (sucrose phosphate synthase) and SUS. Furthermore, sucrose is catalyzed to produce fructose and glucose by INV, SUS, UGP2, pgm, MGAM, and EGLC. This is considered to be the most efficient approach to starch accumulation [73]. SUS is a key enzyme in sucrose metabolism, involved in reversible sucrose decomposition and synthesis, starch synthesis, N fixation, and stress responses [74]. SPS catalyzes Fruc-6P to produce sucrose phosphate, and then it is dephosphorylated to produce sucrose, while INV plays a key role in promoting the decomposition of sucrose. In this research, the majority of the genes mentioned above were upregulated significantly in RCK vs. R13, RCK vs. MRCK, and MRCK vs. MR13 (Figure 8D). In particular, the expression levels of the INV and GBE1 genes were higher than the other comparison group with B13 inoculation after mowing. This indicated that the ability to convert sucrose into glucose and fructose was increased by inoculation after mowing, resulting in the accumulation of the respiratory substrate, promoting root growth and shoot regrowth [75]. It also showed an increase in sucrose catabolism after B13 inoculation under mowing compared with either inoculation or mowing alone, which aligns with a significant accumulation of sucrose content; therefore, strain B13 may further regulate the sucrose metabolism in *L. chinensis* after mowing.

Changes in environmental conditions can affect nitrogen content and disrupt the balance between carbon and nitrogen metabolism in plants [76]. The application of B13 appears to regulate the transcription levels of the NR gene, a critical enzyme in nitrogen metabolism within plant cells, potentially due to the bacterial utilization of NR or the effect of PGPR inoculation on plant photosynthesis and metabolic processes [75]. The downregulation of NR and NiR genes after mowing or inoculation may be related to a feedback regulation mechanism; inoculation may promote nitrate assimilation, potentially leading to reduced enzyme activity [77]. Additionally, after stress events such as mowing, slowed root growth in *L. chinensis* due to lower nitrate absorption from the soil results in the downregulation of N-related genes, including NR and NiR [78]. This suggests that B13 inoculation after mowing promotes nitrogen assimilation. Ammonia (NH_3_) released by plant root absorption, nitrate reduction, photorespiration, and amino acid catabolism is converted to glutamine and glutamate under the action of GS and GOGAT [79]. Furthermore, the upregulation of the GS gene glnA and the GOGAT gene GLT1, the key genes responsible for nitrogen assimilation and remobilization in plants, suggests that plants regulate nitrogen metabolism after mowing or B13 inoculation, especially observed in RCK vs. MR13 (Figure 8B). On the other hand, the increase in GS and GOGAT expression due to inoculation may also be linked to enhanced plant growth [80] and improved stress resilience [65], consistent with the findings in PGPR-treated plants under both control and stress conditions. In summary, B13 inoculation optimizes nitrogen metabolism in *L. chinensis*, especially under mowing stress, by promoting ammonia assimilation through the GS/GOGAT pathway and reducing the demand for nitrate reduction, leading to enhanced plant growth, recovery, and stress resilience.

## 4. Materials and Methods

### 4.1. Identification and Growth-Promoting Properties of B13

This strain was taken from a collection of isolates previously isolated from *L. chinensis* rhizosphere soils in degraded grassland due to long-term overgrazing of the Inner Mongolia Grassland Ecosystem Research Station (43°38′30″ N, 116°42′20″ E) in northern China. The bacterial strain B13 was isolated using the gradient dilution method [38] and preserved in the China General Microbiological Culture Collection Center (CGMCC) under accession number 31,908. The 16S rRNA gene of B13 was amplified from its genomic DNA using universal bacterial primers 27F (AGAGTTTGATCCTGGCTCAG) and 1492R (GGTTACCTTGTTACGACTT). Isolates were identified through UCLUST with a 98.65% similarity threshold, enabling precise species distinction [81]. B13 was stored at −80 °C in LB media containing 30% glycerol.

The qualitative and quantitative assays were conducted to analyze the plant growth-promoting traits of the 13 strains. The qualitative phosphate-solubilizing capacity of B13 was assessed by incubation on Pikovskaya agar mixed with tricalcium phosphate Ca_3_(PO_3_)_2_. The quantitative inorganic phosphate-solubilizing capacity was assessed based on the stannous chloride method with modifications [82]. The IAA production capacity of B13 was assessed as described in Chrastil (1976) [83]. The nitrogenase activity was detected using a double-antibody detection method involving an enzyme antibody (nitrogenase) and a horseradish peroxidase (HRP)-labeled antibody as described by Wang et al. (2024) [84].

### 4.2. Plant Materials and Treatment

*L. chinensis* seeds (West Ujumuqin Leymus, provided by Grassland Research Institute, Chinese Academy of Agricultural Sciences) were soaked in 75% ethanol for 2 min, followed by a 10 min soak in 2% NaClO solution, and then thoroughly rinsed. The uniformly sized seeds of *L. chinensis* were germinated in a culture dish for 10 days. After the seedlings grew to the trefoil stage, they were transplanted into plastic pots (height, 20 cm; diameter, 18 cm) containing approximately 3 kg of field soil, substrate soil, and vermiculite (4:1:1 volume ratio) for 30 days. There were eight seedlings per pot, and each treatment contained six pots in a full factorial design (2 mowing levels × 2 inoculation levels × 6 replicates = 24 pots). The pots were arranged randomly in a greenhouse with a 16/8 h light/dark cycle, day/night temperatures of 30 °C/18 °C, and a relative humidity of 60–70%. Their positions were randomized every three days. Soil moisture was maintained at 60% water-holding capacity. The plants were randomly divided into four groups: non-inoculated and non-mowed control plants (BCK), B13-inoculated plants (B13), non-inoculated mowed plants (MCK), and B13-inoculated mowed plants (M13). After growing for two weeks, half of the *L. chinensis* plants were mowed (stubble height, 6 cm), while the others grew normally. For the inoculation experiment, B13 cells from the LB medium were centrifuged at 8000× *g* for 5 min, after which the supernatant was discarded. The cells were then washed and resuspended in sterile distilled water to be used as the inoculum treatment. The B13 suspension was adjusted to approximately OD_600_ = 1.0 (10^8^ CFU mL^−1^). After mowing over one week, the *L. chinensis* plant roots were inoculated with 10 mL (10^8^ CFU mL^−1^) of B13 suspension (by adding it to the soil around each plant root using a syringe) (B13 and M13) or left with non-inoculated sterile water (BCK and MCK). Only once, 80 mL (10^8^ CFU mL^−1^) of B13 suspension was inoculated in each pot. To assess the growth rate of *L. chinensis* under mowing and non-mowing conditions, the plant height was measured every other week. Four weeks after B13 inoculation, measurements were taken for plant height, stem diameter, leaf width, leaf length, fresh shoot weight, and fresh root weight. In different treatments, the whole roots of two pots of *L. chinensis* were mixed into one sample with a total of three samples used as biological replicates, and immediately frozen and stored at –80 °C for physiological indices, and transcriptome and metabolome evaluation.

### 4.3. Determination of Physiological Indices

The activity of MDA, SOD, PPO, CAT, LOX, and POD, and the bicinchoninic acid (BCA) protein concentration in the root samples from the different treatment groups was measured using corresponding assay kits (Suzhou Keming Biotechnology Co., Ltd., Suzhou, China) [85]. The content of soluble sugar, glucose, sucrose, fructose glucose, TC, and TP were measured using the corresponding assay kits by Suzhou Keming Biotechnology Co., Ltd. Approximately 0.3 g of dried plant material was ground into a fine powder, and the TN and TK contents were measured using a graphite digester (S402, Shandong, China). The ABA, IAA, JA, GA3, and SA levels in fresh plant samples were analyzed using liquid chromatography–tandem mass spectrometry (LC–MS/MS) (Rigol L3000, Beijing, China) [86].

### 4.4. Transcriptome Sequencing and Quantitative PCR Analyses

In this study, 12 root samples were divided into three biological replicates that were used for transcriptome sequencing. Total RNA was extracted from the *L. chinensis* roots using a TRIzol reagent kit (Invitrogen, Carlsbad, CA, USA) according to the manufacturer’s instructions. The RNA integrity and concentration were assessed using a Qubit and Nanodrop (both from Thermo Fisher Scientific, Waltham, MA, USA). The samples with the highest RNA quality were selected for sequencing. Poly(A)-tailed RNAs were purified using oligo(dT) magnetic beads and fragmented into short sequences for cDNA synthesis. The resulting library fragments were purified with a QiaQuick PCR extraction kit (Qiagen, Venlo, The Netherlands). Ligation products were size-selected by agarose gel electrophoresis, amplified by PCR, and sequenced on the Illumina Novaseq X Plus by Gene Denovo Biotechnology Co. (Guangzhou, China). Clean paired-end reads were obtained after raw data processing, filtering, and quality control. Based on the transcriptomic results, several genes were selected for qRT-PCR validation. The primers are listed in Appendix A. 

### 4.5. Metabolomic Profiling of L. chinensis Roots

Freeze-dried root samples of *L. chinensis* were pulverized with zirconia beads using a mixer mill (Retsch GmbH, Düsseldorf, Germany) in liquid nitrogen. Metabolites were extracted at 4 °C with 80% methanol in water, followed by centrifugation for 20 min to remove solids; filtered; and stored in an injection vial for UPLC–MS/MS analysis by Gene Denovo Biotechnology Co., Ltd. (Guangzhou, China). Briefly, 2 μL of each sample was injected into an XSelect HSS T3 column (2.1 × 150 mm, 2.5 μm) at a flow rate of 0.4 mL/min. MS analysis involved electrospray ionization and the following parameters: source temperature at 550 °C, curtain gas at 35 psi, medium collision gas, ion spray voltages at −4500 V and 5500 V, and ion source gases 1 and 2, both at 60 psi. Multiple reaction monitoring (MRM) was applied based on an in-house database. HPLC-MS/MS data files were integrated and peak-corrected using SCIEX OS v1.4 [87].

### 4.6. Statistical Analysis

Statistically significant differences in phenotypic and physiological data were conducted by one-way ANOVA via Tukey’s HSD test in SPSS Statistics (version 19.0, SPSS, Chicago, IL, USA). The results illustrated are the mean value ± standard error. The correlation network between the DEGs and DEMs was analyzed using the “Hmisc” package in R (version 4.02) and visualized in Cytoscape. Figures were created using Origin 2022 (version 9.9), Adobe Illustrator 2022 (version 26.0), and the OmicShare platform (https://www.omicshare.com/; accessed on 6 August 2024).

## 5. Conclusions

This study illustrates how PGPR B13 promotes shoot regrowth in *L. chinensis* after mowing. After mowing, plants prioritize resource allocation to the aboveground parts for recovery, which suppresses root growth, thus reducing nutrient absorption capacity. The plants’ defense response is enhanced, characterized by increased levels of hormones related to defense and damage in the roots, higher antioxidant enzyme activity, and the activation of stress hormone signaling pathways such as JA, SA, and ABA. After mowing, inoculation with PGPR strain B13 enhances root growth and the absorption and utilization of nutrients, which promotes the expression of the genes related to nitrate transport, phosphate transport, and sugar transport, providing sufficient nutrient support for shoot regeneration. It also reduces the levels of JA, SA, and ABA, downregulating the expression of the genes associated with stress response such as ABF, JAZ, and NPR1, and upregulates the signaling of growth-related hormones, helping the plant optimize both growth and defense tradeoffs. PGPR inoculation comprehensively regulates energy metabolism, amino acid metabolism, and carbohydrate metabolism, forming an integrated adaptive mechanism that helps plants more effectively cope with and recover from the damage caused by mowing and improves plant adaptability and regenerative capacity. In summary, applying PGPR after mowing can improve plant resilience and stress resistance in multiple ways, promoting regeneration and biomass accumulation. This approach has broad application potential in grassland management and forage production, optimizing grassland resource utilization, increasing yield, enhancing stress resistance, and promoting the sustainability of grassland ecosystems. We will focus on the long-term impacts of B13 inoculation on L. chinensis regrowth and ecosystem stability in the future.

## Figures and Tables

**Figure 1 ijms-26-00565-f001:**
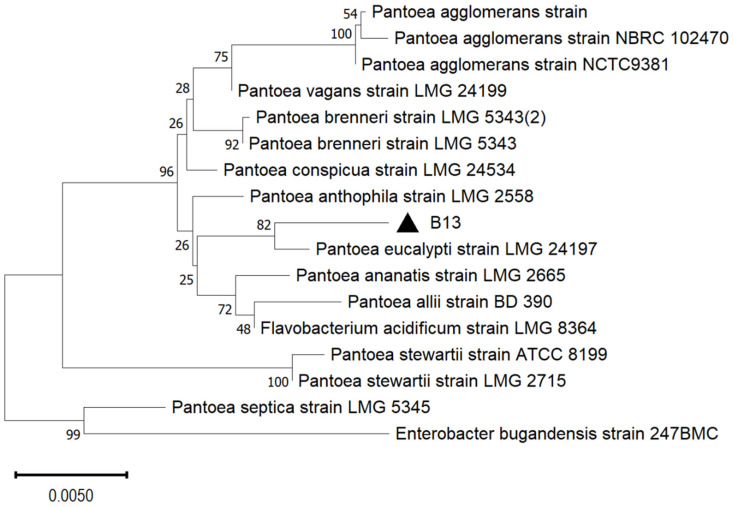
Phylogenetic tree constructed from the 16S rRNA gene sequence analysis showing the isolated strain B13 (marked with a triangle) alongside closely related species from the NCBI GenBank database. This tree was generated using the neighbor-joining method in the MEGA software version 11, with the bootstrap confidence values displayed at each node. The scale bar represents 0.005 nucleotide substitutions per site.

**Figure 2 ijms-26-00565-f002:**
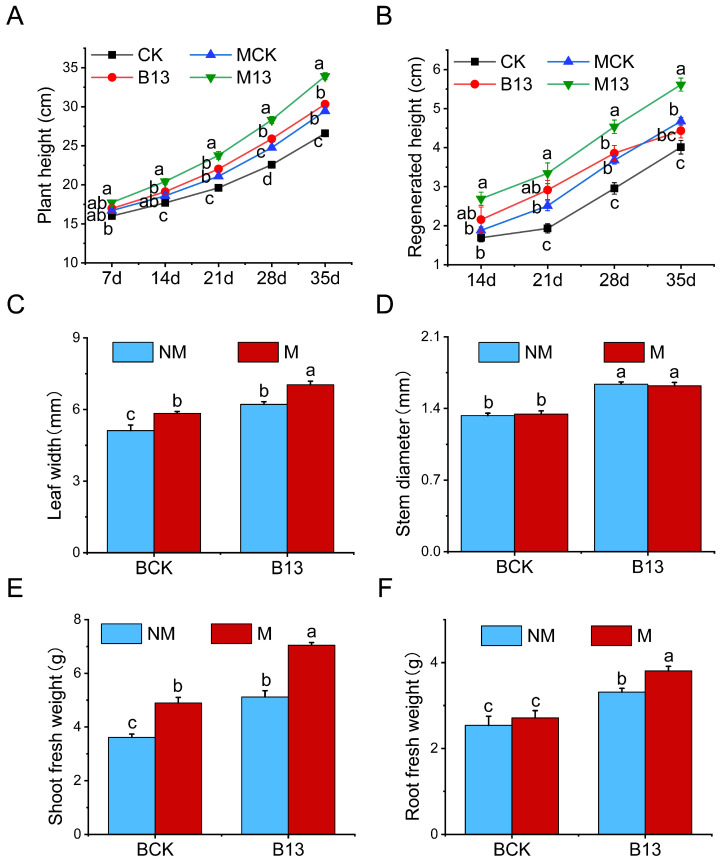
Effect on the growth of *L. chinensis* after B13 inoculation and/or mowing. (**A**) Changes in plant height at 7 d, 14 d, 21 d, 28 d, and 35 d. (**B**) Regrowth height of *L. chinensis* after 14 d, 21 d, 28 d, and 35 d. (**C**–**F**) Changes in the leaf width, stem thickness, shoot fresh weight, and root fresh weight. BCK, control; B13, inoculation with B13; M, mowing; NM, non-mowing. The bar chart shows mean ± SE. Different lowercase letters indicate significant differences between treatments (*p* < 0.05).

**Figure 3 ijms-26-00565-f003:**
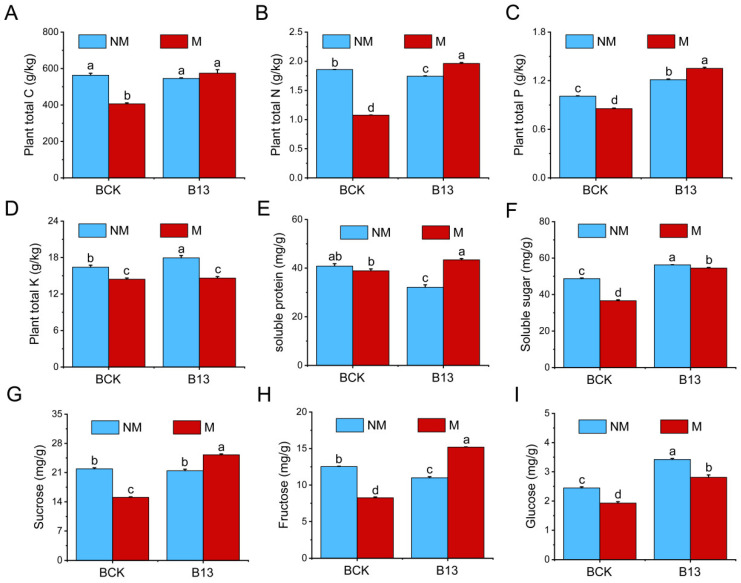
Changes in nutrients and sugar content in *L. chinensis* roots after B13 inoculation and/or mowing. (**A**–**I**) Plant total carbon, plant total nitrogen, plant total phosphorus, plant total potassium, soluble protein, soluble sugar, sucrose, fructose, and glucose. The bar chart shows mean ± SE. Different lowercase letters indicate significant differences between treatments (*p* < 0.05).

**Figure 4 ijms-26-00565-f004:**
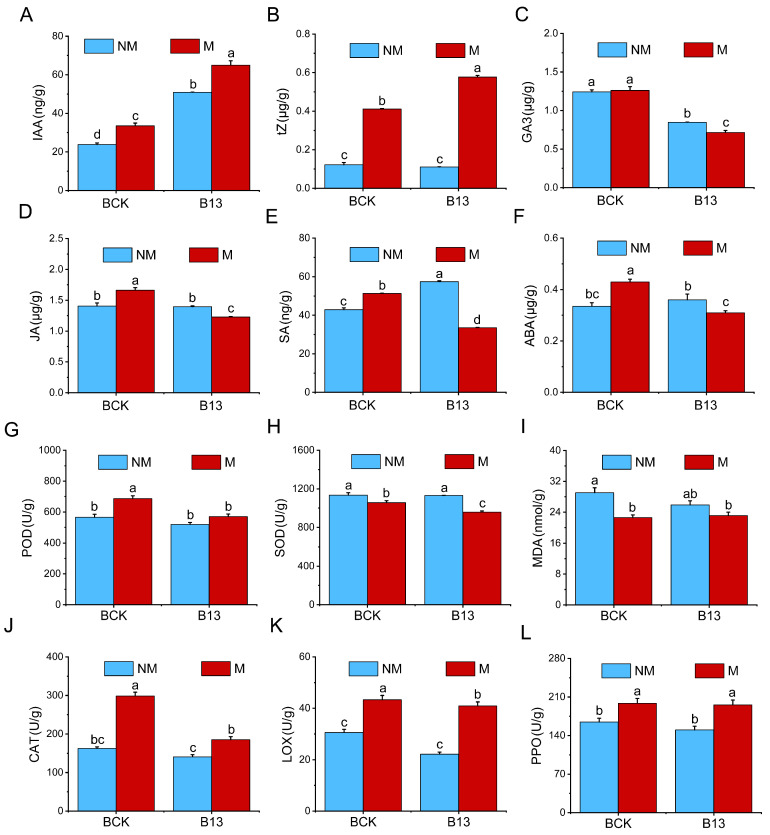
Changes in phytohormone contents and antioxidant enzyme activities in *L. chinensis* roots after B13 inoculation and/or mowing. (**A**–**F**) Indole-3-acetic acid (IAA), zeatin (tZ), gibberellin (GA3), jasmonic acid (JA), salicylic acid (SA), and abscisic acid (ABA); (**G**–**L**) peroxidase (POD, superoxide dismutase (SOD), malondialdehyde (MDA), catalase (CAT), lipoxygenase (LOX), and polyphenol oxidase (PPO). The bar chart shows mean ± SE. Different lowercase letters indicate significant differences between treatments (*p* < 0.05).

**Figure 5 ijms-26-00565-f005:**
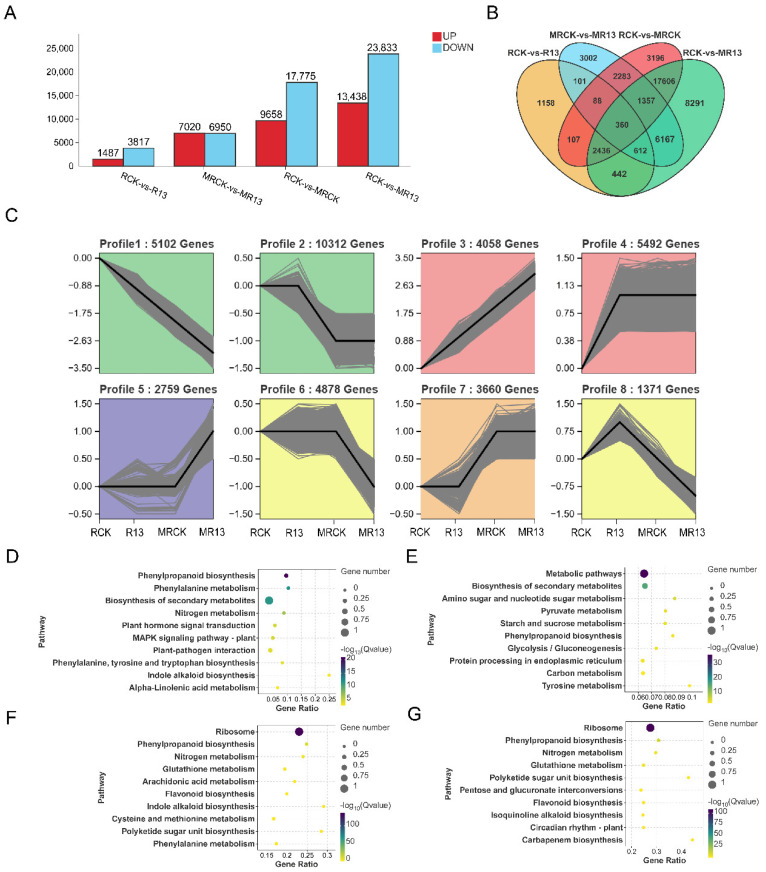
Changes in DEG expression in *L. chinensis* root after inoculation with B13 under mowing or non-mowing. (**A**) Numbers of DEGs in the non-inoculated and B13-inoculated *L. chinensis* root after mowing. Red represents the number of upregulated DEGs, while blue indicates the number of downregulated DEGs. (**B**) Venn diagram of the DEGs showing the genes that are commonly and uniquely expressed across different treatments. (**C**) Expression patterns of the DEGs in the eight significantly enriched profiles, with colors indicating statistical significance for the upregulated and downregulated genes (*p* ≤ 0.05). (**D**–**G**) KEGG pathway enrichment analysis of the DEGs in RCK vs. R13, MRCK vs. MR13, RCK vs. MRCK, and RCK vs. MR13. RCK, control; R13, inoculation with B13; MRCK, mowing; MR13, inoculation with B13 after mowing.

**Figure 6 ijms-26-00565-f006:**
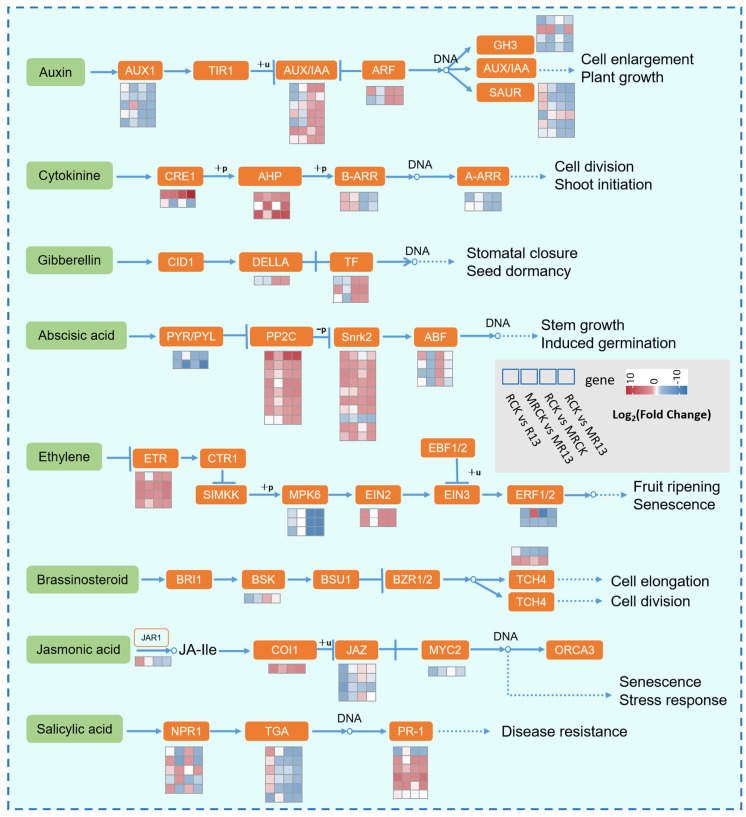
Regulatory network of genes involved in plant hormone signal transduction.

**Figure 7 ijms-26-00565-f007:**
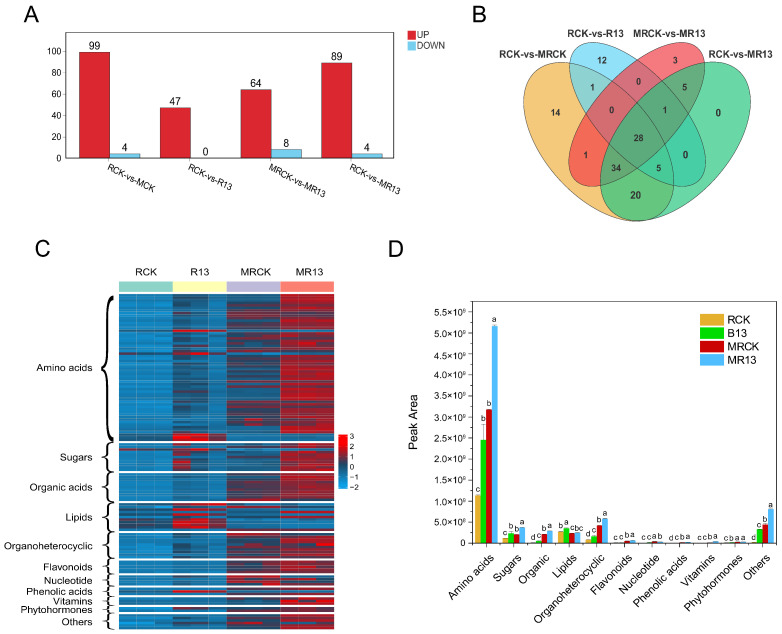
Changes in root DEMs in *L. chinensis* after inoculation with B13 under mowing or non-mowing. (**A**) Numbers of DEMs in the non-inoculated and B13-inoculated *L. chinensis* root after mowing. Red represents the number of upregulated DEMs, while blue indicates the number of downregulated DEMs. (**B**) Venn diagram of the DEMs showing the genes that are commonly and uniquely expressed across different treatments. (**C**) Heatmap of root DEM abundances that differed significantly in the four treatment comparison groups. (**D**) Peak area (abundance) of DEM categories. The bar chart shows mean ± SE. Different lowercase letters indicate significant differences between treatments (*p* < 0.05).

**Figure 8 ijms-26-00565-f008:**
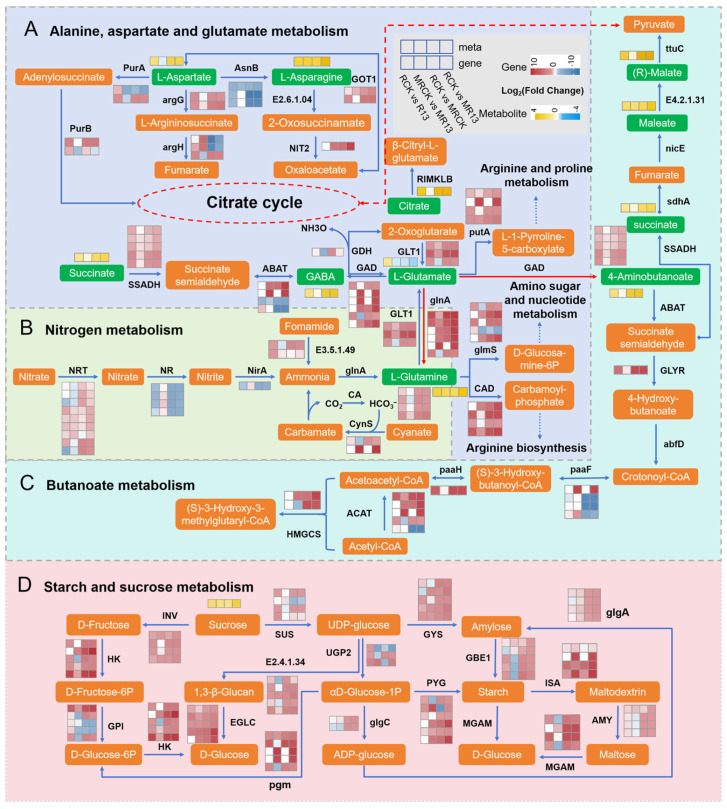
Regulatory network of key genes involved in alanine, aspartate, and glutamate metabolism (**A**); nitrogen metabolism (**B**); butanoate metabolism (**C**); and starch and sucrose metabolism (**D**).

## Data Availability

The original research data is available upon request. All the raw sequencing data have been submitted to the National Center for Biotechnology Information (NCBI) Sequence Read Archive (SRA) database under the accession number PRJNA1177986.

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
