# Peer review of "Combined Transcriptomics and Metabolomics Uncover the Potential Mechanism of Plant Growth-Promoting Rhizobacteria on the Regrowth of Leymus chinensis After Mowing"

_ijms, 2025, doi:10.3390/ijms26020565_

Round 1
Reviewer 1 Report
Comments and Suggestions for Authors
a paper with a clearly stated goal
comments are made with sticky notes
one request is that because methods are last please give more information in Intro and in the results. Please indicate that most of the assays are with root tissue only
It is unclear the degree of replication of the assays - pooling of material is noted but was there only one study and only one metabolome study etc if so although in a single study PCR confirmed transcriptomic results does not mean that the findings are strengthened by repetition
like the fact that this work actually has relevence

Reviewer 2 Report
Comments and Suggestions for Authors
This study explores the effects of plant growth-promoting rhizobacteria (PGPR), specifically Pantoea eucalyptus B13, on the regrowth of Leymus chinensis after mowing. The research combines transcriptomics and metabolomics to understand how PGPR influence nutrient uptake, hormone regulation, and stress resilience in L. chinensis. The results indicate that B13 inoculation enhances plant regrowth by optimizing nutrient absorption, maintaining hormone homeostasis, and promoting metabolic pathways related to carbohydrates and amino acids. The findings have practical applications in agriculture, particularly in enhancing grassland productivity and sustainability. I have some questions and suggestions that should be included before publication.
Q1. The experiments were conducted over a relatively short period (up to 35 days). What are the long-term impacts of B13 inoculation on L. chinensis regrowth and ecosystem stability?
Q2. How applicable are these findings to other grassland species or different environmental conditions?
Minor comments:
Keywords should be arranged alphabetically and not repeat words from the title.
Explain the meaning of statistical symbols in graph footnotes.
What was the number of replications used?
What are the suggestions for future studies (Conclusions)?
Follow the formatting style of MDPI in the Reference List.
After making all the necessary corrections, the manuscript can be accepted for publication.
